energy

plasma, catalyst, carbon dioxide, methanation

**Authors for correspondence:**
Tao He
e-mail: hetao@qibebt.ac.cn
Jinhu Wu
e-mail: wujh@qibebt.ac.cn

This article has been edited by the Royal Society of Chemistry, including the commissioning, peer review process and editorial aspects up to the point of acceptance.

# Plasma-assisted CO₂ methanation: effects on the low-temperature activity of an Ni–Ce catalyst and reaction performance

Yuanzheng Ge[1,2], Tao He[1], Dezhi Han[4], Guihua Li[3], Ruidong Zhao[1] and Jinhu Wu[1]

[1]Key Laboratory of Biofuels, Qingdao Institute of Bioenergy and Bioprocess Technology, Chinese Academy of Sciences, Qingdao 266101, People's Republic of China
[2]University of Chinese Academy of Sciences, Beijing 100049, People's Republic of China
[3]Key Laboratory of Low-Carbon Conversion Science and Engineering, Shanghai Advanced Research Institute, Chinese Academy of Sciences, Shanghai 201210, People's Republic of China
[4]College of Chemical Engineering, Qingdao University of Science and Technology, Qingdao 266042, People's Republic of China

YG, 0000-0002-9908-3568

Ni–Ce three-dimensional material with macropore diameter of 146.6 ± 8.4 nm was synthesized and used as a methanation catalyst. Firstly, H₂ reduction of the catalyst was conducted in the thermal fixed bed and plasma reactor, respectively, then X-ray diffraction (XRD) and CO₂ temperature programmed desorption experiments on the two reduced samples were carried out to reveal the plasma effect on the catalyst's physico-chemical properties. It was found that plasma reduction created more abundant basic sites for CO₂ adsorption, in particular the medium basic sites were even doubled compared with the thermal-reduced catalysts. The plasma-reduced catalyst exhibited excellent low-temperature activity, *ca* 50–60°C lower than the thermal catalyst (the maximum CO₂ conversion point). Based on the optimum reduced catalyst, plasma effect in the reactor level was further investigated under high gas hour space velocity of approximately 50 000 h⁻¹. The plasma reactor showed higher CO₂ conversion capacity and efficiency than the thermal reactor.

## 1. Introduction

Catalytic methanation of CO₂ was first discovered by Sabatier in 1902 [1,2]. Restrained by the lack of cheap hydrogen sources,

few of the methanation technologies reached the commercial level [1]. Due to the increasing demand to store the surplus renewable power, the concept of $CO_2$ methanation coupled with water electrolysis, known as power to gas (PtG), was proposed by German in 2009. This concept revived the ancient technology and brought new attention to it [1,3]:

$$CO_2(g) + 4H_2(g) \rightarrow CH_4(g) + 2H_2O(g), \quad \Delta H_{298K} = -165 \text{ kJ mol}^{-1}. \tag{R1}$$

Apparently, reaction (1) is a highly exothermic and volume contrast reaction, which means pressure and temperature significantly influence the reaction equilibrium. From the thermodynamic viewpoint, it is desirable to operate the reaction at low temperature to achieve high $CO_2$ conversion. However, low temperature also means slow reaction kinetics and when the temperature is below 250°C, the conversion is negligible. At temperature above 450°C, the $CH_4$ yield decreases significantly with the CO rapid formation as a by-product through the reversed water gas shift reaction route [4–6]. Also, high temperature usually leads to the sinter of the catalyst.

In recent years, numerous studies have been conducted on the catalysts of $CO_2$ methanation. Among all the catalysts, Ni-based catalysts are the most widely investigated material catalysing the hydrogenation of $CO_2$ to methane due to their relatively high activity and low cost. These studies focused on improving the activity, stability and the resistance to carbon deposition of the catalysts by adding promoters, loading on different supports and modifying catalyst preparation methods [1–3,7–14]. Ceria is often employed as a promoter to improve Ni nanoparticles dispersion and decrease the size; besides, it can also work as a single support. Oxygen vacancies in ceria caused by the existence of $Ce^{3+}/Ce^{4+}$ ion pairs lead to efficient activation of $CO_2$ [7,8,12]. Nie et al. [8] found that the addition of $CeO_2$ could enhance the catalyst activity by lowering Ni particle size and generating oxygen vacancies. The equilibrium conversion can be achieved over Ni–Ce supported $\gamma$-$Al_2O_3$ catalyst at 300°C with optimal 3wt% $CeO_2$ addition.

Plasma known as a partially ionized state consists of many active species such as electrons, ions and radicals. Non-thermal plasma (NTP) can generate high energetic electrons (1–10 eV), while the bulk temperature can keep as low as room temperature [15–17]. Among the NTP technologies, the dielectric barrier discharge (DBD) plasma is convenient to couple with the catalyst bed. The combination of catalyst with DBD plasma could initiate chemical reactions such as ionization, excitation and dissociation at a lower temperature and may also have an additional effect on the physico-chemical properties of the catalyst. Nizio et al. [6,18] reported methanation with 80% $CO_2$ conversion and nearly 100% $CH_4$ selectivity under the assistance of plasma. Wang et al. [19] prepared a highly dispersed Co nanoparticle catalyst and created more active sites using a plasma-assisted method. Besides, the quick on–off switch property of DBD methanation equipment is suitable for the PtG system.

Moreover, the catalyst's structural properties such as pore volume, pore size distribution and surface area are critical to $CO_2$ methanation activity [2,7,8,12]. Three-dimensionally ordered macroporous (3DOM) material possesses ordered array and uniform size of macropores larger than 50 nm with walls formed by nanoparticles. The unique pore structure of 3DOM materials allows gases to diffuse from all directions to the pores, thus reducing the mass transport resistance [20–30].

In this work, the 3DOM Ni–Ce catalyst was successfully synthesized and a comparative study for the catalytic performance of the 3DOM catalyst was evaluated both in the DBD plasma reactor and thermal fixed-bed reactor. This well-fabricated catalyst exhibited an excellent catalytic activity and high selectivity for methane under high gas hour space velocity (GHSV) with the assistance of plasma.

# 2. Experimental procedure

## 2.1. Catalyst preparation

All reagents purchased from Sinopharm Chemical Reagent Co., Ltd. are analytical-grade, apart from the methyl methacrylate which is chemical-grade, and they were used as received without further purification. The 3DOM Ni–Ce catalyst was synthesized through Ni and Ce precursor solutions co-impregnating on the poly-methyl methacrylate (PMMA) colloidal crystal template.

### 2.1.1. Synthesis of PMMA colloidal crystal template

Monodisperse PMMA spheres were prepared via a soap-free emulsion polymerization method. Typically, the polymerization took place in a four-neck round-bottom flask equipped with a thermometer, an inlet for $N_2$ feeding, a water-cooling condenser and a mechanical stirring paddle. First, 580 ml deionized water and

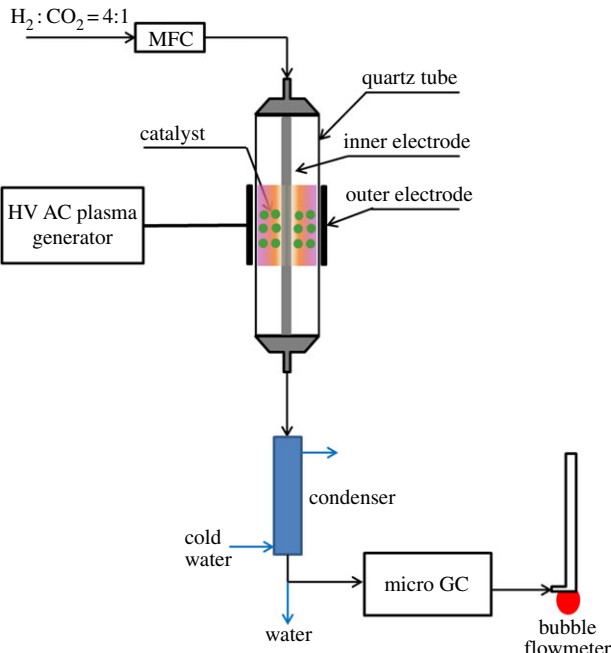

**Figure 1.** Schematic of the catalytic-plasma reactor.

24.48 g methyl methacrylate were added into the flask. The solution was heated to 70°C by a hot water bath under constant stirring at 350 r.p.m. and $N_2$ bubbling. In a separate 50 ml beaker, 0.20 g potassium persulfate (KPS) was dissolved in 20 ml deionized water and then also heated to 70°C. After $N_2$ bubbling to remove oxygen for at least 30 min, the preheated KPS solution was added to the flask, and then the polymerization was allowed to run at 70°C for 65 min under constant stirring and $N_2$ bubbling. The resulting monodisperse PMMA microsphere emulsion was cooled to room temperature and then centrifuged at 2200 r.p.m. for 6 h. After abandoning the supernatant liquor, the solid matter was dried at 40°C, and the highly ordered PMMA colloidal crystal template was obtained [20,22,27,28].

### 2.1.2. Synthesis of 3DOM Ni–Ce catalyst

Prior to the impregnation, the PMMA template was heated to 100°C for 5 min in the oven to increase the mechanical strength and eliminate the remaining MMA in the template. Typically, 5.04 g $Ce(NO_3)_3 \cdot 6H_2O$, and 3.87 g $Ni(NO_3)_2 \cdot 6H_2O$ (molar ratio 1.15 : 1) were dissolved in 25.0 ml 95% ethanol solution with the addition of equimolar citric acid (5.30 g) as a complexing agent. PMMA template was impregnated with a continuous droplet of the Ni–Ce precursors on a Buchner funnel under vacuum condition. Then, the infiltrated template was removed from the funnel and transferred to a vacuum oven. It was dried at 40° for 4 h and then dried at 60°C for another 4 h. These impregnation and drying steps were repeated four times to fill the template uniformly and completely. The hybrids were heated to 300°C at a ramp rate of 1°C min$^{-1}$, and held at 300°C for 1 h. Then, the precursor was heated again at 1°C min$^{-1}$ to 550°C and calcined for 6 h. After cooling down, the 3DOM Ni–Ce catalyst was obtained [21,22,25,30].

## 2.2. Catalyst test and analysis

### 2.2.1. Experimental set-up and procedure

$CO_2$ methanation over a 3DOM Ni–Ce catalyst were evaluated in two reactors: a DBD plasma reactor and a thermal fixed-bed reactor. The schematic diagram of the DBD plasma reactor is shown in figure 1. The DBD plasma reactor consists of a quartz tube as the dielectric material, stainless rod as the inner electrode linked to the cover and ground, stainless net outside the tube as the high-voltage electrode. The plasma discharges in the tube gap between the electrodes. The 0.375 g catalyst (30–60 mesh) diluted with 2.00 g quartz sand of the same size were loading in the plasma zone. The voltage and current applied to the plasma system is 55–60 V and 1.55–1.60 A during methanation reaction, respectively, and the output voltage is *ca* 20 kV.

The thermal catalytic methanation of $CO_2$ was performed in a vertical continuous flow fixed-bed quartz tube surrounded by a temperature-controlled electrical oven. The same amount of catalysts and

quartz sand were placed in the middle of the tube and furnace. The reaction temperature was measured by a K-type thermocouple penetrated into the catalyst bed.

The gas injection was the same for the two reactors controlled by the calibrated mass flow controller (MFC). A gas mixture of $H_2$ and $CO_2$ with a molar ratio of 4 : 1 was fed into the reactor for methanation reaction. The effluent gases were cooled down in a condenser at −5.0°C and passed through a drier to remove the water. Then, the dried gases were analysed by an online gas chromatograph (GC) equipped with TCD and the flow rate was also monitored by a bubble flowmeter. The influence factor for each component was calibrated by two standard gas mixtures through the external standard method. The overall mass balance on carbon basis was more than 98%. Thus, the conversion of $CO_2$ ($X_{CO_2}$) and the selectivity of $CH_4$ ($S_{CH_4}$) and CO ($S_{CO}$) were calculated as follows:

$$X_{CO_2} = \frac{F_{CH_4} + F_{CO}}{F_{CO_2} + F_{CH_4} + F_{CO}} \times 100\%, \tag{2.1}$$

$$S_{CH_4} = \frac{F_{CH_4}}{F_{CH_4} + F_{CO}} \times 100\% \tag{2.2}$$

and

$$S_{CO} = \frac{F_{CO}}{F_{CH_4} + F_{CO}} \times 100\%, \tag{2.3}$$

where $F_i$ were the molar fraction of $i$ component ($i = CH_4$, CO and $CO_2$) in the effluent gases.

### 2.2.2. Catalyst reduction method

Prior to the catalytic methanation, the catalysts need to be reduced to the active phases under $H_2$ atmosphere. Two reduction methods were investigated for comparison: plasma reduction and thermal reduction. The plasma-assisted reduction was conducted in the DBD reactor under 100 ml min$^{-1}$ hydrogen and plasma atmosphere with 70 V and 1.70 A electrical input for 30 min. The thermal reduction was performed in the thermal fixed-bed reactor under 100 ml min$^{-1}$ $H_2$, and three temperatures (350, 400 and 450°C) were chosen for comparison with a retention time of 30–60 min.

## 2.3. Characterization

Scanning electron microscope (SEM) analyses of 3DOM catalyst were performed on a HITACHI SU8010 operating at 10.0 kV to investigate the morphology and structure of the catalyst.

$N_2$ isothermal adsorption–desorption analysis was conducted on a Micromeritics ASAP 2020 Sorptometer at −196°C. Prior to the analysis, the sample was degassed at 200°C for 6 h with a ramping rate of 10°C min$^{-1}$. The specific surface area was calculated by the Brunauer–Emmett–Teller (BET) method. The sample had a macroporous structure and the Barrett–Joyner–Halenda (BJH) method is only appropriate for mesoporous materials. The density functional theory (DFT) method was employed to obtain the pore size distribution.

Powder X-ray diffraction (XRD) patterns of catalysts were measured with a Bruker D8 Advance diffractometer using Cu–K$_\alpha$ radiation at 40 kV and 40 mA at a scanning rate of 4° min$^{-1}$ over the $2\theta$ range of 5–80° with a step size of 0.02°. The crystallite sizes of metallic Ni, NiO (200) and $CeO_2$ were calculated by the Scherrer equation based on Ni (111) and $CeO_2$ (111) peaks, respectively.

$CO_2$ temperature programmed desorption (TPD) was performed on a Micromeritics Chemisorption Analyzer 2920TR equipped with a thermal conductivity detector (TCD). The catalyst sample was first treated in He flow at 300°C for 120 min with a ramping rate of 10°C min$^{-1}$. After cooling down to 100°C, the sample was exposed to a 10% $CO_2$/He gas mixture with a flow rate of 20 ml min$^{-1}$ for 90 min. Then the sample was swept by 20 ml min$^{-1}$ He for 80 min to remove the physically absorbed $CO_2$ until the baseline remained stable. Finally, the signal of TCD was recorded from 100 to 650°C with a heating rate of 10°C min$^{-1}$.

# 3. Results and discussion

## 3.1. Morphology and pore size of catalyst

The morphology and macrostructure of PMMA colloidal crystal template and Ni–Ce catalyst were observed using SEM technology, as shown in figure 2. The PMMA colloidal template is uniform and

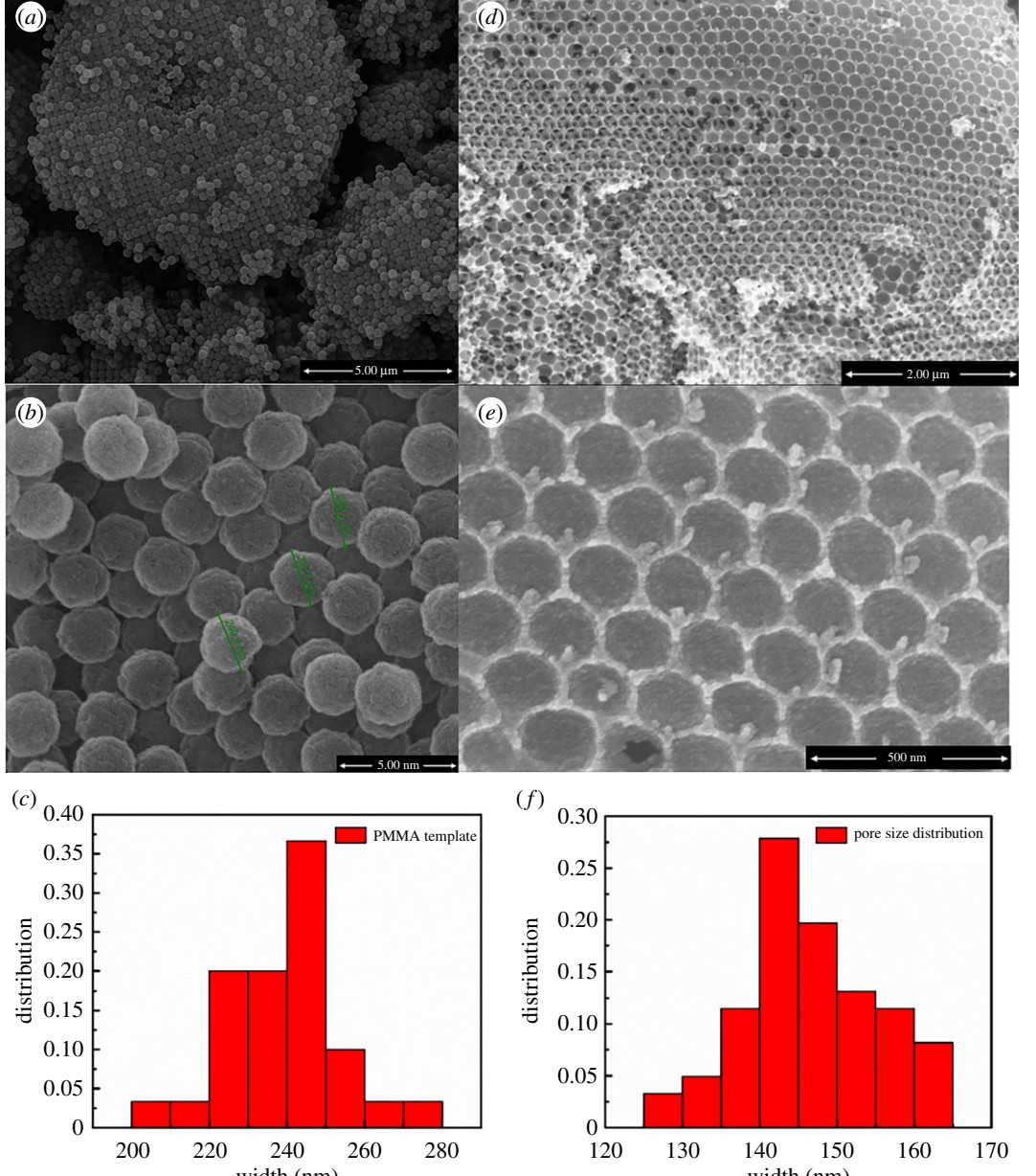

**Figure 2.** SEM images of the PMMA template and 3DOM Ni–Ce catalyst. (*a–c*) PMMA template; (*d–f*) 3DOM NiO–CeO$_2$ catalyst.

ordered over a large scope with the existence of both simple cubic packing and cubic close packing (figure 2*a*), although there also exist some conglomeration defects. The monodispersed PMMA in the template is spherical with a diameter of 239.5 ± 15.0 nm (figure 2*b,e*). From figure 2*c,d* and *f*, it can be seen that the NiO–CeO$_2$ material is an inverse replica of the PMMA template, possessing open and ordered 3D macropores with an inner diameter of 146.6 ± 8.4 nm. Compared with the PMMA colloidal template, the shrinkage of the pores is about 39%, which was caused by the PMMA sphere shrinkage during the calcination. And this is consistent with the 32% contraction ratio of 3DOM Al$_2$O$_3$ derived from the PMMA colloidal template [22] and 60% contraction ratio of 3DOM CeO$_2$ derived from the polystyrene template [21].

The N$_2$ isothermal adsorption and desorption and the pore size distribution are shown in figure 3. The physisorption isotherm exhibits a type V with an H3 hysteresis loop which indicates the existence of macropores. The BET surface is 32.19 m$^2$ g$^{-1}$ in consistence with 30–40 m$^2$ g$^{-1}$ for 3DOM CeO$_2$ [21]. The DFT method for pore size distribution reveals that macropores larger than 100 nm contribute a lot to the pore volume and there also exists a certain number of micropores of around 3 nm, which may be caused by the calcination of citric acid.

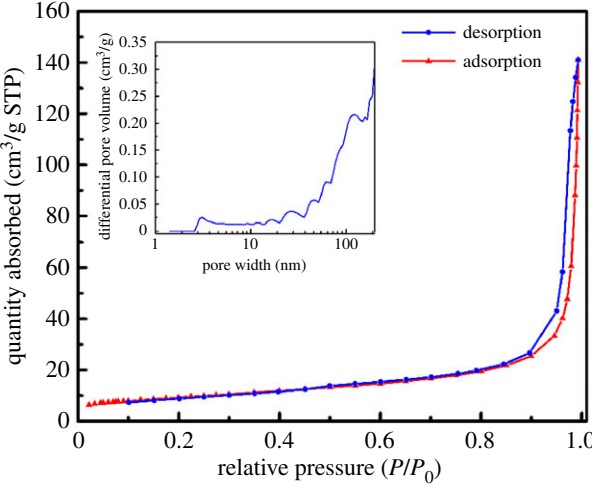

**Figure 3.** $N_2$ isothermal adsorption–desorption of 3DOM Ni–Ce catalyst samples.

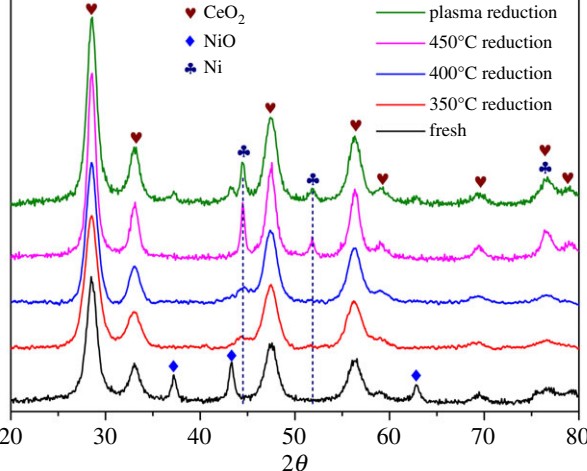

**Figure 4.** XRD patterns of the Ni–Ce catalyst samples with different reduction conditions.

## 3.2. Effect of $H_2$-reduction conditions on catalyst's chemical state and $CO_2$ adsorption characteristics

$H_2$ reduction is an important pre-activation process for the Ni-related catalysts. Different reduction conditions may lead to different crystal size, chemical state and defect sites. Plasma-assisted $H_2$ reduction is different from thermal reduction, which experiences a mild temperature field (200–250°C) coupled with plenty of high energetic electrons and H radicals. To investigate the effect of plasma on the reduction process, catalyst reduced under plasma was compared with the thermal-reduced samples, in terms of XRD and $CO_2$-TPD characteristics. And in order to eliminate random error, three thermal reduction temperatures (350,400° and 450°C) were chosen for comparison

The XRD patterns including the fresh catalyst are shown in figure 4, the peaks at $2\theta = 28.5, 33.1, 47.5, 56.3, 58.9, 69.3, 76.7$ and 79.0 correspond to the $CeO_2$ phase; the peaks at $2\theta = 37.2, 43.3$ and 62.8 represent the NiO phase; the peaks at $2\theta = 44.5, 51.8$ and 76.4 reflect the existence of metallic Ni. Apparently, NiO and $CeO_2$ are the main phases in fresh catalysts. Plasma reduction and 450°C thermal reduction yield clear and sharp Ni peaks, while a minor NiO phase still exists in the plasma-reduced catalyst. A similar phenomenon was also detected by Tu *et al*. [31] in $H_2$/Ar plasma-assisted reduction of Ni/$Al_2O_3$ catalyst. And Zhang *et al*. [32] found that the $H_2$-TPR result of Ni/$SiO_2$ catalyst after $N_2$ and $H_2$ plasma treatment demonstrated the existence of unreduced NiO phase. During the plasma-assisted reduction process, the temperature was less than 250°C, and the incomplete reduction of NiO may be caused by the mild temperature condition. There are only irregular and small drum-bags at the

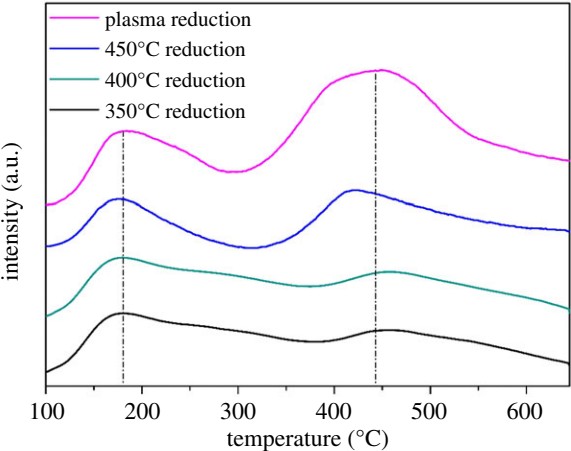

**Figure 5.** $CO_2$-TPD profiles of catalysts after plasma and thermal reduction.

positions of $2\theta = 44.5$ for the catalysts reduced under 350 and 400°C. This indicates that NiO is reduced slowly and does not form the regular Ni crystal phase under lower thermal temperatures.

The chemical adsorption of $CO_2$ on the catalyst is the first step in the methanation reaction and plays a significant role in the activation process [6,12]. The $CO_2$ adsorption properties can be evaluated by $CO_2$-TPD experiment. The $CO_2$-TPD results of catalyst samples reduced in plasma and thermal conditions, respectively, are illustrated in figure 5. Apparently, there are two $CO_2$ desorption peaks over the temperature range investigated: one is around 175°C and another is around 450°C. The first peak can be ascribed to the existence of weak basic sites such as OH and O groups, and the second peak is due to medium basic sites such as $O^{2-}$ [2]. Obviously, the order of $CO_2$ desorption peak area is: 350°C reduction $\approx$ 400°C reduction < 450°C reduction << plasma reduction. The normalized peak area of weak basic sites is 12.4 for the plasma-reduced catalyst, while it is 9.1 for the 450°C thermal-reduced one; the peak area of medium basic sites in the plasma-reduced catalyst is more than double that of the 450°C thermal-reduced catalyst (33.6 versus 14.7).

Compared to Ni–Ce catalysts reported in the literature [6], the present Ni–Ce catalyst reduced under plasma has more abundant medium basic sites, due to the second peak area being much larger than that of the first peak. According to the literature [33], during the methanation, $CO_2$ chemisorption on basic sites forms carbonate and it is then hydrogenated to methane via an intermediate-formate species. $CO_2$ adsorption on the weak basic sites such as OH and O groups tends to form hydrogen carbonate that becomes bidentate carbonate during further hydrogenation. It is prone to form monodentate carbonate and is then hydrogenated to monodentate formate, if the $CO_2$ adsorption occurs on the medium basic sites such as the $O^{2-}$ group [33]. The further hydrogenation of monodentate formate takes place faster than that of bidentate carbonate. The medium basic sites play a more significant role than the weak sites for this kind of catalyst in enhancing the catalytic activity during $CO_2$ methanation [33]. Thus, plasma reduction could produce more abundant basic sites, especially the medium basic sites in the catalyst, which may favour higher $CO_2$ conversion capacity and efficiency.

## 3.3. Effect of plasma on reaction performance

### 3.3.1. Effect of plasma reduction on catalytic activity

Methanation reaction is highly exothermic, low temperature is thermodynamically favourable for high $CO_2$ conversion and the development of catalysts possessing low-temperature activity has attracted much attention. In §3.2, it was found that plasma reduction brought more medium basic sites and mild plasma reduction may lead to smaller crystal size representing higher activity. Therefore, low-temperature activity of different catalyst samples reduced in the plasma and thermal reactor was tested in the thermal reactor by regulating the furnace temperature gradually, as shown in figure 6. Due to its exothermal character, once the trigger-point of methanation reaction is reached, a temperature-rise in bed temperature will be detected by the thermocouple inserted in the middle of the catalyst bed. Therefore, $CO_2$ conversion is plotted versus furnace temperature and bed temperature, respectively, in figure 6 to give a comprehensive view of the activity test.

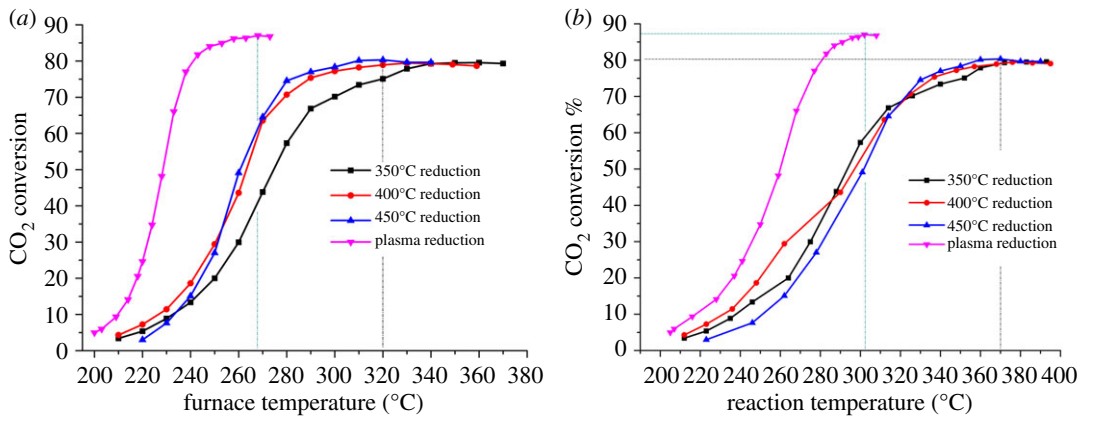

**Figure 6.** Activity test curves of 3DOM Ni–Ce catalyst reduced under different conditions, (*a*) $CO_2$ conversion versus furnace temperature, (*b*) $CO_2$ conversion versus catalyst bed temperature; reaction condition: thermal fixed-bed reactor, GHSV = 8600 h$^{-1}$, $H_2 : CO_2 = 4 : 1$.

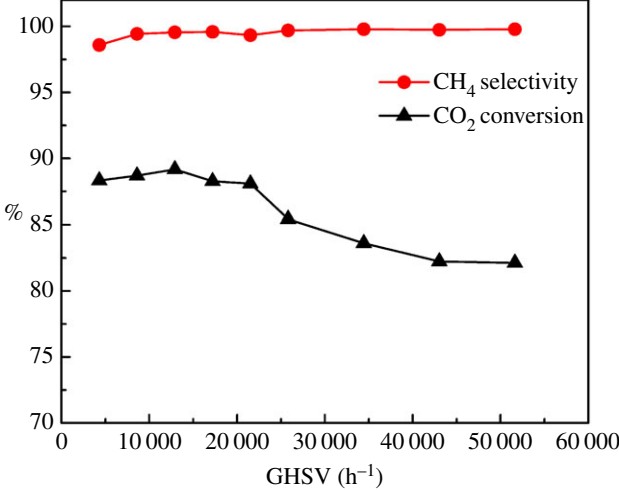

**Figure 7.** The influence of GHSV on methanation performance in the thermal reactor. Reaction condition: furnace temperature 210–225°C, bed temperature 430–525°C, $H_2 : CO_2 = 4 : 1$, 3DOM Ni–Ce catalyst reduced in the plasma reactor first, then transferred to the thermal reactor for methanation after cooling down to room temperature.

Obviously, the catalyst reduced under plasma conditions showed a higher low-temperature activity and $CO_2$ conversion than those reduced by the thermal method. As shown in figure 6*a*, the trigger temperature of the plasma-reduced catalyst is in the range of 190–200°C, and when the furnace temperature reached 268°C, the $CO_2$ conversion reached the maximum *ca* 87%. On the other hand, the trigger temperature of the thermal-reduced catalyst is 15–20°C higher than that of the plasma catalyst, and the maximum $CO_2$ conversion was only *ca* 80% until the furnace temperature reached *ca* 320°C. Figure 6*b* also shows a similar temperature difference in the catalyst bed. The maximum $CO_2$ conversion over plasma-reduced catalyst corresponds to a bed temperature of *ca* 303°C, and it is *ca* 370°C for the case of thermal-reduced catalysts. The CH$_4$ selectivity is almost the same for the two cases around 99.4%. For the thermal reduction process, it is clear that 450°C is an optimum condition, which yields clear Ni crystal peaks in the XRD spectrum. Furthermore, it is found that the order of the catalyst's activity is in accordance with the capacity of $CO_2$ adsorption (shown in figure 5); in particular, it is more related to the quantity of the medium basic sites.

### 3.3.2. Effect of plasma on conversion capacity

Besides high activity, more efficient conversion of $CO_2$ in per unit reactor or catalyst is also an important factor. In order to investigate the feasibility and capacity of the plasma reactor, high GHSV experiments were carried out both in the thermal and plasma reactor with the same catalyst samples for comparison. Figure 7 shows the results of the GHSV test in the thermal fixed-bed reactor loading with optimal Ni–Ce catalyst (plasma reduced). As shown, when the GHSV increased from 8600 to 40 000 h$^{-1}$, although the

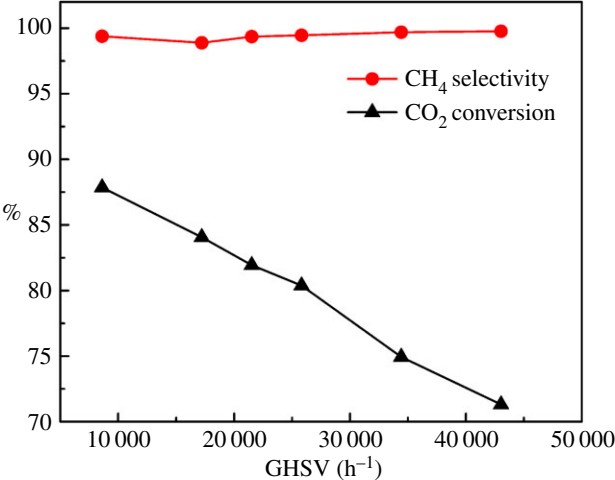

**Figure 8.** The influence of GHSV on methanation performance in the plasma reactor. Reaction condition: input 57 V and 1.56 A, $H_2 : CO_2 = 4 : 1$, 3DOM Ni–Ce catalyst reduced in the plasma reactor.

$CH_4$ selectivity kept almost constant, the $CO_2$ conversion drastically and linearly decreased from 87.8 to 71.3%. At the same time, it was found that the catalyst bed temperature ramped up from 430 to 525°C with the increase in GHSV, due to reaction heat.

Figure 8 shows the contrast experiment results from the DBD plasma reactor. Apparently, the $CO_2$ conversion curve in figure 8 is flatter than the one in figure 7, and the plasma reactor showed higher conversion capacity and efficiency for methanation reaction. In detail, the $CO_2$ conversion remained constant between 88 and 89% in the GHSV range of 4000–20 000 h$^{-1}$ and then a slow descent occurred at GHSV of 25 000 h$^{-1}$; but the downward trend gradually disappeared, and the conversion maintained at *ca* 82% from 40 000 to 50 000 h$^{-1}$ GHSV. The high conversion capacity of the DBD plasma reactor should be due to the unique method of molecular activation through high energetic electrons (1–10 ev). The electrons can activate molecules of $CO_2$ and $H_2$ in a more directed and efficient manner. The low temperature of plasma atmosphere will also lead to a low adiabatic reaction temperature that thermodynamically favours $CO_2$ conversion.

## 4. Conclusion

Firstly, the 3DOM Ni–Ce catalyst with pore diameter of 146.6 ± 8.4 nm was successfully synthesized through inverse replica of PMMA templates, and it was combined with plasma technology to enhance the efficiency and capacity of the $CO_2$ methanation reaction. In the catalyst reduction process, plasma demonstrated a positive impact on the material's physico-chemical properties. $CO_2$-TPD results revealed that the plasma-reduced catalyst possessed more abundant weak and medium basic sites than the thermal-reduced catalysts, favouring $CO_2$ adsorption and activation. A methanation reaction test in the thermal fixed-bed reactor showed that plasma reduction generated higher low-temperature activity than the thermal method. Furthermore, the plasma reaction mode was compared with the thermal mode. The $CO_2$ conversion decreased almost linearly with the increase in GHSV when used in the thermal reactor. Instead, the plasma reactor gave a relatively stable and higher $CO_2$ conversion capacity; even GHSV reached 50 000 h$^{-1}$, $CO_2$ conversion was still 82%. This work contributes to a further understanding of the plasma's effect on the catalyst's characteristics and reaction capacity.

Ethics. Our investigation was carried out in full accordance with the ethical guidelines of our research institution and in compliance with Chinese legislation.

Data accessibility. Data supporting this study are available from the Dryad Digital Repository: https://doi.org/10.5061/dryad.kg76k5b [34].

Authors' contributions. Y.G. prepared the catalysts, conducted the experiment and wrote the manuscript; T.H. designed and guided the experiment, and amended the manuscript; D.H. prepared the catalyst and guided the experiment; G.L. cooperated on this project; R.Z. characterized the catalysts; J.W. administrated the project. All authors gave final approval for publication.

Competing interests. We declare we have no competing interests.

Funding. T.H. acknowledges the support of the Foundation of Key Laboratory of Low-Carbon Conversion Science & Engineering, Shanghai Advanced Research Institute, Chinese Academy of Sciences (grant no. KLLCCSE-201704,

SARI, CAS) and the National Key R&D Program of China (grant no. 2017YFE0105500). D.H. would like to thank the Start-up Foundation for Advanced Talents of Qingdao University of Science and Technology (grant no. 010022919). Acknowledgements. We thank Miss Xinxin Liu and Miss Shanshan Zhu for the assistance with experiments and also thank Dr Zhuo Li for her guidance in catalysts preparation and helpful suggestions.

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
