## [Reviewer comments · Royal Society Open Science]

Review History

RSOS-190750.R0 (Original submission)

Review form: Reviewer 1

Is the manuscript scientifically sound in its present form?

No

Are the interpretations and conclusions justified by the results?

No

Is the language acceptable?

Yes

Is it clear how to access all supporting data?

Yes

Do you have any ethical concerns with this paper?

No

Have you any concerns about statistical analyses in this paper?

No

Recommendation?

Major revision is needed (please make suggestions in comments)

Comments to the Author(s)

In this work, Ge and coworkers presented an experimental investigation on the synthesis and catalytic performance characterization of three dimensionally ordered mesoporous materials as catalyst for CO₂ methanation. The authors attributed the superior performance of the plasma reduced catalyst to the small Ni CeO₂ crystallites and abundant basic sites for CO₂ adsorption. By carefully reading, I found the following should be addressed before the manuscript can be considered further.

1. For CO₂ methanation, both the reactants and the products are among the smallest molecules, would the conversion and selectivity sensitive to the surface area and pore size?
2. On Page 10, in the middle, the authors claimed that "A minor NiO phase still existed in the plasma reduced catalysts. The incomplete reduction of NiO may be caused by the mild reduction condition." I don't think it sound. Please address it further in the context, or put citations to relevant experimental reports to justify.
3. Further, for determination of crystallite size of Ni and CeO₂, more details on the calculation of the crystallite size should be shown in the context. A citation should be put to the Scherrer Equation. In fact, I think both the peak position and peak area of Ni and CeO₂ crystallite don't differ significantly. This makes it interesting to discuss the reliability of the calculated crystallite size. I strongly suggest the authors to do a high resolution transmission electronic microscopy characterization to show the exact size of the crystallite size.
4. The authors claimed at the end of page 10 "The smaller crystal sizes imply that plasma assisted reduction led to a better dispersion of the active phase in the catalysts.....". I think these proposals need further evidence.
5. On page 11, while discussing the CO₂ adsorption, the authors mentioned that formate will be formed during the reaction. However, I didn't see any evidence for these, at least a citation should be made here to justify the proposal. Also, the desorption of CO₂ at 180 degree C and 430 degree C is special. How are these values as compared with the adsorption of CO₂ on relevant Ni/CeO₂ catalysts? Further, I suggest the author to check the composition of the reduced catalyst to confirm that there is no C in the reduced catalysts.

Review form: Reviewer 2**Is the manuscript scientifically sound in its present form?**

No

Are the interpretations and conclusions justified by the results?

Yes

Is the language acceptable?

Yes

Is it clear how to access all supporting data?

Yes

Do you have any ethical concerns with this paper?

No

Have you any concerns about statistical analyses in this paper?

Yes

Recommendation?

Accept with minor revision (please list in comments)

Comments to the Author(s)

The authors synthesised a macroporous Ni/CeO₂ for CO₂ methanation. The catalyst was activated by reduction under thermal conditions (under H₂ at 470C) and under plasma (H₂ under 20 kV). The activities of the two catalysts were compared under thermal reactor conditions. The better performance of the plasma-reduced catalyst was ascribed to the smaller nanoparticles of Ni and CeO₂ on the surface and the higher number of basic adsorption sites of CeO₂ for the plasma-reduced catalyst. The performance of the plasma-reduced catalyst was evaluated under thermal and plasma conditions. The plasma reactor was better able to convert CO₂ at high flow rates.

The manuscript is interesting and would probably be of interest to the community. The language is mostly acceptable, with a few grammatical issues that I imagine will be picked up by the editors.

The main scientific issue I have with this work is the way that the authors have chosen to demonstrate the superiority of plasma activation of the catalyst over thermal activation. Only one set of conditions (each) is used for plasma and thermal, and the result is two different catalysts. That one is better than the other is almost a certainty and cannot necessarily be ascribed to plasma being the better method: It might be that a better catalyst could be produced by thermal activation under a different set of conditions. For the thermal catalyst, the Ni(II) appears to be fully reduced, according to the XRD, while the plasma catalyst still has some Ni(II). It may be that the thermal method not only thoroughly reduced the Ni(II), but continued long enough to produce other changes, such as an increase in particle sizes through annealing and atomic migration. If the thermal reduction had been stopped at an earlier time, or had been performed at a lower temperature, perhaps the resulting, thermally-activated catalyst would have been more comparable to those in the plasma-activated catalyst. It might be that the authors optimised the two activation processes. If so, they should describe the optimisation and provide data to demonstrate that the activated catalysts were optimised.

Minor changes:

On page 3, the authors state that the plasma has a "synergetic effect on the physicochemical properties of catalysts." Although the concept of synergy (synergistic or synergetic) is used very loosely in the literature, it should only be applied to situations in which the adjustment of two variables together creates a much larger effect than changing one variable and then the other in sequence. That was not shown in this paper. I would call the effect of the plasma on the catalyst surface an "additional" effect.

On page 4 the authors should add detail about their synthetic method. For example, what were the concentrations of Ce, Ni and citric acid in their ethanol solution? Was the infiltrated template removed from the funnel during the drying stages, or was the entire funnel and template dried together? The line, "The hybrids were calcined at 550C for 6 h with a hold at 300C for 1 h at a ramp rate of 1 C/min" is a bit confusing. I would suggest, "The temperature of the hybrids was

raised to 300C at a rate of 1 C/min, held at 300C for 1 h, raised again at 1 C/min to 550C, and calcined at that temperature for 6 h."

On page 6, the authors should describe how they calibrated the GC to obtain mole fraction values.

On page 8, the figure would be easier to read if the authors stacked the panels for the precursor template on the left and the panels for the catalysts on the right.

On page 10 the authors claim that the crystallites of Ni and CeO₂ are smaller for the plasma treatment, but do not provide confidence intervals.

The authors should consider tweaking the legends in their figures to help readers who print the article in black and white: Organise the legends so that the top legend line matches the top trace, etc.

Decision letter (RSOS-190750.R0)

25-Jun-2019

Dear Dr Ge:

Title: Plasma assisted CO₂ methanation over a 3DOM catalyst: effects on physicochemical properties of catalyst and reaction performance

Manuscript ID: RSOS-190750

The editor assigned to your manuscript has now received comments from reviewers. We would like you to revise your paper in accordance with the referee and Subject Editor suggestions which can be found below (not including confidential reports to the Editor). Please note this decision does not guarantee eventual acceptance.

Please submit your revised paper before 18-Jul-2019. Please note that the revision deadline will expire at 00.00am on this date. If we do not hear from you within this time then it will be assumed that the paper has been withdrawn. In exceptional circumstances, extensions may be possible if agreed with the Editorial Office in advance. We do not allow multiple rounds of revision so we urge you to make every effort to fully address all of the comments at this stage. If deemed necessary by the Editors, your manuscript will be sent back to one or more of the original reviewers for assessment. If the original reviewers are not available we may invite new reviewers.

When submitting your revised manuscript, you must respond to the comments made by the referees and upload a file "Response to Referees" in "Section 6 - File Upload". Please use this to

document how you have responded to the comments, and the adjustments you have made. In order to expedite the processing of the revised manuscript, please be as specific as possible in your response.

RSC Associate Editor:
Comments to the Author:
(There are no comments.)

RSC Subject Editor:
Comments to the Author:
(There are no comments.)

Reviewers' Comments to Author:
Reviewer: 1

Comments to the Author(s)

In this work, Ge and coworkers presented an experimental investigation on the synthesis and catalytic performance characterization of three dimensionally ordered mesoporous materials as catalyst for CO₂ methanation. The authors attributed the superior performance of the plasma reduced catalyst to the small Ni CeO₂ crystallites and abundant basic sites for CO₂ adsorption. By carefully reading, I found the following should be addressed before the manuscript can be considered further.

1. For CO₂ methanation, both the reactants and the products are among the smallest molecules, would the conversion and selectivity sensitive to the surface area and pore size?
2. On Page 10, in the middle, the authors claimed that "A minor NiO phase still existed in the plasma reduced catalysts. The incomplete reduction of NiO may be caused by the mild reduction condition." I don't think it sound. Please address it further in the context, or put citations to relevant experimental reports to justify.
3. Further, for determination of crystallite size of Ni and CeO₂, more details on the calculation of the crystallite size should be shown in the context. A citation should be put to the Scherrer Equation. In fact, I think both the peak position and peak area of Ni and CeO₂ crystallite don't differ significantly. This makes it interesting to discuss the reliability of the calculated crystallite

size. I strongly suggest the authors to do a high resolution transmission electronic microscopy characterization to show the exact size of the crystallite size.

4. The authors claimed at the end of page 10 “The smaller crystal sizes imply that plasma assisted reduction led to a better dispersion of the active phase in the catalysts.....”. I think these proposals need further evidence.

5. On page 11, while discussing the CO₂ adsorption, the authors mentioned that formate will be formed during the reaction. However, I didn't see any evidence for these, at least a citation should be made here to justify the proposal. Also, the desorption of CO₂ at 180 degree C and 430 degree C is special. How are these values as compared with the adsorption of CO₂ on relevant Ni/CeO₂ catalysts? Further, I suggest the author to check the composition of the reduced catalyst to confirm that there is no C in the reduced catalysts.

Reviewer: 2

Comments to the Author(s)

The authors synthesised a macroporous Ni/CeO₂ for CO₂ methanation. The catalyst was activated by reduction under thermal conditions (under H₂ at 470C) and under plasma (H₂ under 20 kV). The activities of the two catalysts were compared under thermal reactor conditions. The better performance of the plasma-reduced catalyst was ascribed to the smaller nanoparticles of Ni and CeO₂ on the surface and the higher number of basic adsorption sites of CeO₂ for the plasma-reduced catalyst. The performance of the plasma-reduced catalyst was evaluated under thermal and plasma conditions. The plasma reactor was better able to convert CO₂ at high flow rates.

The manuscript is interesting and would probably be of interest to the community. The language is mostly acceptable, with a few grammatical issues that I imagine will be picked up by the editors.

The main scientific issue I have with this work is the way that the authors have chosen to demonstrate the superiority of plasma activation of the catalyst over thermal activation. Only one set of conditions (each) is used for plasma and thermal, and the result is two different catalysts. That one is better than the other is almost a certainty and cannot necessarily be ascribed to plasma being the better method: It might be that a better catalyst could be produced by thermal activation under a different set of conditions. For the thermal catalyst, the Ni(II) appears to be fully reduced, according to the XRD, while the plasma catalyst still has some Ni(II). It may be that the thermal method not only thoroughly reduced the Ni(II), but continued long enough to produce other changes, such as an increase in particle sizes through annealing and atomic migration. If the thermal reduction had been stopped at an earlier time, or had been performed at a lower temperature, perhaps the resulting, thermally-activated catalyst would have been more comparable to those in the plasma-activated catalyst. It might be that the authors optimised the two activation processes. If so, they should describe the optimisation and provide data to demonstrate that the activated catalysts were optimised.

Minor changes:

On page 3, the authors state that the plasma has a “synergetic effect on the physicochemical properties of catalysts.” Although the concept of synergy (synergistic or synergetic) is used very loosely in the literature, it should only be applied to situations in which the adjustment of two variables together creates a much larger effect than changing one variable and then the other in sequence. That was not shown in this paper. I would call the effect of the plasma on the catalyst surface an “additional” effect.

On page 4 the authors should add detail about their synthetic method. For example, what were the concentrations of Ce, Ni and citric acid in their ethanol solution? Was the infiltrated template removed from the funnel during the drying stages, or was the entire funnel and template dried together? The line, "The hybrids were calcined at 550C for 6 h with a hold at 300C for 1 h at a ramp rate of 1 C/min" is a bit confusing. I would suggest, "The temperature of the hybrids was raised to 300C at a rate of 1 C/min, held at 300C for 1 h, raised again at 1 C/min to 550C, and calcined at that temperature for 6 h."

On page 6, the authors should describe how they calibrated the GC to obtain mole fraction values.

On page 8, the figure would be easier to read if the authors stacked the panels for the precursor template on the left and the panels for the catalysts on the right.

On page 10 the authors claim that the crystallites of Ni and CeO₂ are smaller for the plasma treatment, but do not provide confidence intervals.

The authors should consider tweaking the legends in their figures to help readers who print the article in black and white: Organise the legends so that the top legend line matches the top trace, etc.

Author's Response to Decision Letter for (RSOS-190750.R0)

See Appendix A.

RSOS-190750.R1 (Revision)

Review form: Reviewer 1

Is the manuscript scientifically sound in its present form?

Yes

Are the interpretations and conclusions justified by the results?

Yes

Is the language acceptable?

Yes

Do you have any ethical concerns with this paper?

No

Have you any concerns about statistical analyses in this paper?

No

Recommendation?

Accept as is

Comments to the Author(s)

In this revision, the authors have addressed most of my concerns and the manuscript may be publishable in its current form.

Review form: Reviewer 2

Is the manuscript scientifically sound in its present form?

No

Are the interpretations and conclusions justified by the results?

No

Is the language acceptable?

Yes

Do you have any ethical concerns with this paper?

No

Have you any concerns about statistical analyses in this paper?

No

Recommendation?

Major revision is needed (please make suggestions in comments)

Comments to the Author(s)

The authors have addressed the minor points I made in my previous review; however, I don't believe that they understood the point I was trying to make in what I describe as my "main scientific issue."

It seems to me that the two research questions this paper is mean to address are (1) whether low temperature plasma is the better method for reductively activating the Ni catalysts described and (2) whether plasma is a better method than thermal for methanation. I believe that the latter is better answered than the former because the same catalyst was used in the methanation experiments, so that was controlled, and the conditions (flow rate and temperature) varied during those experiments, so there was a kind of optimisation built in to the methanation experiment.

On the other hand, the only thing the activation experiments demonstrate is that under two different reduction conditions, two different catalysts can be produced from the same starting material. The data do not show that plasma is the better method. The authors have provided references and data to show that their choice of thermal reduction conditions (100 mL/min H₂, 470C, 60 min.) were appropriate to activate the catalyst. That was never in dispute: The thermally-activated catalyst had reasonable activity, although lower than the plasma-activated one. My point is that the question of whether plasma is – in general – the better method for activating these Ni catalysts is not answered, just because one particular plasma-activated catalyst outperformed one particular thermally-activated catalyst.

So what would answer the question? I believe that if the authors attempted to optimise the conditions for the thermally-activated catalyst (reaction time, hydrogen flow and reduction temperature), and set their very best thermally-activated catalyst against their one plasma-activated catalyst, that would be sufficiently convincing. If their unoptimised plasma catalyst can beat their optimised thermal catalyst, I would have no argument.

Decision letter (RSOS-190750.R1)

22-Jul-2019

Dear Dr Ge:

Title: Plasma assisted CO₂ methanation over a 3DOM catalyst: effects on physicochemical properties of catalyst and reaction performance
Manuscript ID: RSOS-190750.R1

The editor assigned to your paper has now received comments from reviewers. We would like you to revise your paper in accordance with the referee and Subject Editor suggestions which can be found below (not including confidential reports to the Editor). Please note this decision does not guarantee eventual acceptance.

Please submit a copy of your revised paper before 14-Aug-2019. Please note that the revision deadline will expire at 00.00am on this date. If we do not hear from you within this time then it will be assumed that the paper has been withdrawn. In exceptional circumstances, extensions may be possible if agreed with the Editorial Office in advance. We do not allow multiple rounds of revision so we urge you to make every effort to fully address all of the comments at this stage. If deemed necessary by the Editors, your manuscript will be sent back to one or more of the original reviewers for assessment. If the original reviewers are not available we may invite new reviewers.

Please also include the following statements alongside the other end statements. As we cannot publish your manuscript without these end statements included, if you feel that a given heading is not relevant to your paper, please nevertheless include the heading and explicitly state that it is not relevant to your work.

- Funding statement

Please include a funding section after your main text which lists the source of funding for each author.

RSC Associate Editor:
Comments to the Author:
(There are no comments.)

RSC Subject Editor:
Comments to the Author:
(There are no comments.)

Reviewers' Comments to Author:
Reviewer: 2

Comments to the Author(s)

The authors have addressed the minor points I made in my previous review; however, I don't believe that they understood the point I was trying to make in what I describe as my "main scientific issue."

It seems to me that the two research questions this paper is mean to address are (1) whether low temperature plasma is the better method for reductively activating the Ni catalysts described and (2) whether plasma is a better method than thermal for methanation. I believe that the latter is better answered than the former because the same catalyst was used in the methanation experiments, so that was controlled, and the conditions (flow rate and temperature) varied during those experiments, so there was a kind of optimisation built in to the methanation experiment.

On the other hand, the only thing the activation experiments demonstrate is that under two different reduction conditions, two different catalysts can be produced from the same starting material. The data do not show that plasma is the better method. The authors have provided

references and data to show that their choice of thermal reduction conditions (100 mL/min H₂, 470C, 60 min.) were appropriate to activate the catalyst. That was never in dispute: The thermally-activated catalyst had reasonable activity, although lower than the plasma-activated one. My point is that the question of whether plasma is –in general– the better method for activating these Ni catalysts is not answered, just because one particular plasma-activated catalyst outperformed one particular thermally-activated catalyst.

So what would answer the question? I believe that if the authors attempted to optimise the conditions for the thermally-activated catalyst (reaction time, hydrogen flow and reduction temperature), and set their very best thermally-activated catalyst against their one plasma-activated catalyst, that would be sufficiently convincing. If their unoptimised plasma catalyst can beat their optimised thermal catalyst, I would have no argument.

Reviewer: 1

Comments to the Author(s)

In this revision, the authors have addressed most of my concerns and the manuscript may be publishable in its current form.

Author's Response to Decision Letter for (RSOS-190750.R1)

See Appendix B.

RSOS-190750.R2 (Revision)

Review form: Reviewer 2

Is the manuscript scientifically sound in its present form?

Yes

Are the interpretations and conclusions justified by the results?

Yes

Is the language acceptable?

Yes

Do you have any ethical concerns with this paper?

No

Have you any concerns about statistical analyses in this paper?

No

Recommendation?

Accept as is

Comments to the Author(s)

The authors have addressed all my concerns.

Decision letter (RSOS-190750.R2)

09-Sep-2019

Dear Dr Ge:

Title: Plasma assisted CO₂ methanation: effects on the low-temperature activity of a Ni-Ce catalyst and reaction performance
Manuscript ID: RSOS-190750.R2

It is a pleasure to accept your manuscript in its current form for publication in Royal Society Open Science. The chemistry content of Royal Society Open Science is published in collaboration with the Royal Society of Chemistry.

RSC Associate Editor:
Comments to the Author:
(There are no comments.)

RSC Subject Editor:
Comments to the Author:
(There are no comments.)

Reviewer(s)' Comments to Author:
Reviewer: 2

Comments to the Author(s)
The authors have addressed all my concerns.

Appendix A

Responses to Referees' comments

Responses to referee 1

Original general comments:

In this work, Ge and coworkers presented an experimental investigation on the synthesis and catalytic performance characterization of three dimensionally ordered mesoporous materials as catalyst for CO₂ methanation. The authors attributed the superior performance of the plasma reduced catalyst to the small Ni CeO₂ crystallites and abundant basic sites for CO₂ adsorption. By carefully reading, I found the following should be addressed before the manuscript can be considered further.

Response: Thanks for the reviewer's comments for improving the quality of this manuscript. In this version, more details and more citation are added to improve the fidelity and readability of our work. Furthermore, more discussions are also added to interpret the results. And we hope that the manuscript could meet the criteria of the *Royal Open Science Society*. The detailed response for each comment is listed below:

Original comment 1:

For CO₂ methanation, both the reactants and the products are among the smallest molecules, would the conversion and selectivity sensitive to the surface area and pore size?

Response:

There is no strong correlation between BET surface area, pore structure and catalyst activity in CO₂ methanation (T. A. Le *et al.*, 2017, *Catal. Today.*, 293-294, 89-96). The reason for the selection of 3DOM material is that the discharge in the catalyst bed may be enhanced (U. Kogelschatz, 2003, *Plasma Chem. Plasma P.*, 23, 1-46).

According to the literature (T. A. Le *et al.*, 2017, *Catal. Today.*, 293-294, 89-96), Ni-CeO₂ with intermediate surface area and micropore structure possesses a better CO₂ methanation activity. While for the Ni-Ce catalyst with largest surface area and well developed mesopores (G. Zhou *et al.*, 2017, *Int. J. Hydrogen Energ.*, 42, 16108-16117) exhibited the best activity. The surface area and pore size do affect the catalyst activity,

but they are not the key factor to determine the catalyst performance. The conversion and selectivity are not sensitive to the surface area and pore size.

In DBD plasma reactor, not only the reactions occurs in the gas-phase and at the catalytic surface, but also the discharge appears between 2 electrodes through the dielectric material and the packed catalyst bed (U. Kogelschatz, 2003, *Plasma Chem. Plasma P.*, 23, 1-46, G. Petitpas *et al.*, 2007, *Int. J. Hydrogen Energ.*, 32, 2848-2867, A. Parastayev *et al.*, 2018, *Appl. Catal. B-Environ.*, 239, 168-177). The unique property of plasma is the way that the molecules are activated by ionization, excitation, and dissociation. The discharge in plasma plays a significant role in initiating the CO₂ methanation. The catalyst with macropores could occupy more volume of the discharge zone and the intensity of discharge could be enhanced because permittivity of catalyst is higher than gas. On other hand, for 3DOM catalyst, the discharge not only appears between particles, but also takes places between the orderly skeleton of the catalyst which leads to a more uniform micro-discharge(U. Kogelschatz, 2003, *Plasma Chem. Plasma P.*, 23, 1-46). And this may enhance the catalyst performance.

Original comment 2:

On Page 10, in the middle, the authors claimed that “A minor NiO phase still existed in the plasma reduced catalysts. The incomplete reduction of NiO may be caused by the mild reduction condition.” I don't think it sound. Please address it further in the context, or put citations to relevant experimental reports to justify.

Response:

During the plasma assisted reduction process, there is no external heat input and the bed temperature was below 250 °C. While, in thermal reduction the bed temperature kept at 470 °C. Compared with the bed temperature of thermal reduction, we called the plasma reduction “mild condition”.

The XRD patterns shows NiO phase exists in plasma reduced catalyst, and the similar results were observed in the literatures: X.Tu (X. Tu *et al.*, 2013, *Catal. Today*, 211, 120-125) noted Ni/Al₂O₃ catalyst was not fully reduced in H₂/Ar plasma-assisted reduction through XRD patterns; Xu, Zhang (X. Zhang *et al.*, 2013, *J. Fuel Chem.*

Techn., 41, 96-101) found that H₂-TPR result of Ni/SiO₂ catalyst after N₂ and H₂ plasma treatment demonstrated the existence of unreduced NiO phase.

Original comment 3:

Further, for determination of crystallite size of Ni and CeO₂, more details on the calculation of the crystallite size should be shown in the context. A citation should be put to the Scherrer Equation. In fact, I think both the peak position and peak area of Ni and CeO₂ crystallite don't differ significantly. This makes it interesting to discuss the reliability of the calculated crystallite size. I strongly suggest the authors to do a high-resolution transmission electronic microscopy characterization to show the exact size of the crystallite size.

Response:

According to the suggestions, more details and interpretations on the calculation of the crystallite size were added to improve the fidelity and readability, as shown in the revised manuscript. And the Scherrer Equation are also cited as Equation 4 (T. A. Le *et al.*, 2017, *Catal. Today.*, 293-294, 89-96). When processing the XRD patterns, all the peaks were carefully identified, fitted and refined to obtain all the profiles. And the residual error of fit was less than 5% for each catalyst. As shown in Table 1, the FWHM of Ni, NiO and CeO₂ differ significantly.

The crystallite size calculated by Scherrer Equation were done to obtain the average size by many researchers. The Scherrer Equation takes all sample into consideration and the reliability is based on the particle shape and inner strains. Measurement and calculation of HRTEM image is another way to obtain the crystallite size. W.Nie (W. Nie *et al.*, 2017, *Fuel*, 202, 135-143) conducted these 2 methods to characterize the Ni crystallite size and both the 2 results showed the NiO–3CeO₂/γ-Al₂O₃ had the smallest Ni crystallite. M. Nizio (M. Nizio *et al.*, 2016, *Catal. Commun.*, 83, 14-17) also use Scherrer Equation to obtain the Ni crystallite size for different catalysts. T.A Le (T. A. Le *et al.*, 2017, *Catal. Today.*, 293-294, 89-96) also use Scherrer Equation to determine Ni crystallite size. Both of the 2 methods are reliable. But we

regretted that this test cannot be conducted due to the availability of the HRTEM equipment in our lab. Therefore, we only supplied the calculated results instead.

Original comment 4:

The authors claimed at the end of page 10 “The smaller crystal sizes imply that plasma assisted reduction led to a better dispersion of the active phase in the catalysts ……” . I think these proposals need further evidence.

Response:

For a certain loading of Ni, the smaller crystallite size comes with more Ni particles which means a better dispersion. It leads that H₂ molecules activated by Ni are closer to CO₂ molecules absorbed on CeO₂ facilitating hydrogenation of CO₂ to methane. S. Abelló (S. Abelló et al., 2013, Fuel, 113, 598-609) reported a Ni-Al catalyst with the smaller Ni crystallite size exhibited a better CO₂ conversion than the other catalysts. Better performance was observed by W. Nie (W. Nie et al., 2017, Fuel, 202, 135-143) on Ni-Ce-Al catalyst with smallest Ni crystallite size. The similar assertions were also made by M. Nizio (M. Nizio et al., 2016, Int. J. Hydrogen Energ., 41, 11584-11592) and Q. Liu (Q. Liu et al., 2017, Int. J. Hydrogen Energ., 42, 12295-12300). Thus, it is general accepted that a smaller crystal size means a better dispersion and activity.

The activity curve of 3DOM catalysts reduced in thermal reactor is shown in Fig. 1 (in context as Fig 6). The reactions occurred at GHSV=8,600 h⁻¹, H₂:CO₂=4:1. For thermal reduced catalyst with bigger crystallite sizes, when the bed temperature reached 290 °C, a slight increase of furnace temperature would ignite the reaction causing a rapid rise of bed temperature to 372 °C, and the CO₂ conversion rose rapidly to 79% simultaneously; therefore, the trigger temperature of methanation on thermal reduced catalysts should be around 290 °C. The maximum CO₂ conversion was ca. 80% with the CH₄ selectivity of 99.4%. For plasma assisted reduced catalyst with smaller crystallite sizes, the trigger temperature of methanation on plasma reduced catalysts was much lower, around 225 °C. Furthermore, the plasma reduced catalysts gave a maximum CO₂ conversion of 87.8% almost equivalent to the thermodynamic

equilibrium result of the same temperature. The 3DOM catalyst reduced in the H₂-plasma atmosphere has smaller crystallites and higher catalytic activity with almost the same CH₄ selectivity.

Figure 1. Activity test curves of Ni-Ce samples in thermal fixed bed reactor; reaction condition: GHSV=8,600 h⁻¹, H₂:CO₂=4:1. (a) the sample was reduced in thermal reactor; (b) the sample was firstly reduced in plasma reactor and then transferred to the thermal reactor for methanation after cooling down to room temperature.

Original comment 5:

On page 11, while discussing the CO₂ adsorption, the authors mentioned that formate will be formed during the reaction. However, I didn't see any evidence for these, at least a citation should be made here to justify the proposal. Also, the desorption of CO₂ at 180 degree C and 430 degree C is special. How are these values as compared with the adsorption of CO₂ on relevant Ni/CeO₂ catalysts? Further, I suggest the author to check the composition of the reduced catalyst to confirm that there is no C in the reduced catalysts.

Response:

The discussion on the role of the medium basic sites plays in CO₂ chemisorption was based on the literatures. In the literature (Q. Pan *et al.*, 2014, *Catal Commun*, 45, 74-78), the significant role of medium basic sites in Ni-Ce catalyst is identified and well-discussed. CO₂ adsorption on the medium basic sites such as O²⁻ group is faster than the other pathways in CO₂ methanation. For CO₂ desorption peak, it varies with

the structure and composition. According to the literature (T. A. Le *et al.*, 2017, *Catal. Today.*, 293-294, 89-96), the Ni-CeO₂ catalyst also shows 2 peaks at around 150 °C and 500 °C. And the Ni-CeO₂ catalyst also shows 2 peaks at around 130 °C and 300 °C in (M. Nizio *et al.*, 2016, *Catal. Commun.*, 83, 14-17). And Ni-Ce-Zr catalyst only shows a peak at 500 °C by (M. Nizio *et al.*, 2016, *Int. J. Hydrogen Energ.*, 41, 11584-11592). After rechecking the XRD and CO₂-TPD data and comparison with literatures, the desorption of CO₂ at 180 °C and 430 °C is reasonable.

Responses to referee 2

Original general comments:

The authors synthesised a macroporous Ni/CeO₂ for CO₂ methanation. The catalyst was activated by reduction under thermal conditions (under H₂ at 470C) and under plasma (H₂ under 20 kV). The activities of the two catalysts were compared under thermal reactor conditions. The better performance of the plasma-reduced catalyst was ascribed to the smaller nanoparticles of Ni and CeO₂ on the surface and the higher number of basic adsorption sites of CeO₂ for the plasma-reduced catalyst. The performance of the plasma-reduced catalyst was evaluated under thermal and plasma conditions. The plasma reactor was better able to convert CO₂ at high flow rates.

The manuscript is interesting and would probably be of interest to the community. The language is mostly acceptable, with a few grammatical issues that I imagine will be picked up by the editors.

Response: Thanks for the reviewer's questions and comments for improving the quality of this manuscript. In this version, more details and citations are added to improve the fidelity of our work. Furthermore, as the referee recommended, figures are modified and rearranged to improve the readability of the manuscript. And we hope that the manuscript could meet the criteria of the **Royal Open Science Society**. The detailed response for each comment is listed below:

Original comment 1:

The main scientific issue I have with this work is the way that the authors have chosen to demonstrate the superiority of plasma activation of the catalyst over thermal activation. Only one set of conditions (each) is used for plasma and thermal, and the result is two different catalysts. That one is better than the other is almost a certainty and cannot necessarily be ascribed to plasma being the better method: It might be that a better catalyst could be produced by thermal activation under a different set of conditions. For the thermal catalyst, the Ni(II) appears to be fully reduced, according to the XRD, while the plasma catalyst still has some Ni(II). It may be that the thermal method not only thoroughly reduced the Ni(II), but continued long enough to produce other changes, such as an increase in particle sizes through annealing and atomic migration. If the thermal reduction had been stopped at an earlier time, or had been performed at a lower temperature, perhaps the resulting, thermally-activated catalyst would have been more comparable to those in the plasma-activated catalyst. It might be that the authors optimised the two activation processes. If so, they should describe the optimisation and provide data to demonstrate that the activated catalysts were optimised.

Response:

Yes, we clearly know the concern of the reviewer on the reduction procedure. Metallic Ni is the main active phase, and the reduction process should occur at an appropriate temperature and last for an appropriate time to ensure fully reduction of NiO and meanwhile avoid the sintering of Ni. The conditions of the reduction process were based on H₂-TPR test, previous activity test and literature review.

Figure 2 H₂-TPR profiles of Ni-Ce catalyst

From the result of H₂-TPR, the reduction takes place in the range of 400~500 °C. From the literature (T. A. Le *et al.*, 2017, *Catal. Today.*, 293-294, 89-96), the thermal reduction took place at 500 °C. In the literature (S. Abelló *et al.*, 2013, *Fuel*, 113, 598-609), the reduction occurred at 500 °C and last for 3 h in 10% H₂/N₂ atmosphere. According to M. Nizio (M. Nizio *et al.*, 2016, *Int. J. Hydrogen Energ.*, 41, 11584-11592), the catalyst was previously reduced in the presence of 5% vol. H₂/Ar mixture at 470 °C for 2 h. The reduction temperature varies with the loading of Ni, structure catalysts and interaction between the NiO and the support. Thus, in our work, the catalyst was reduced under 100 mL/min H₂ at 470 °C for 60 min. And from XRD pattern, the NiO phase is fully reduced to metallic Ni. If the thermal reduction occurs at lower temperature and takes shorter time, un-fully reduced NiO phase and poor performance will be observed.

For plasma assisted reduction, according to literature (X. Tu *et al.*, 2013, *Catal. Today*, 211, 120-125), Ni/Al₂O₃ catalyst was not fully reduced in H₂/Ar plasma-assisted process. To ensure the higher reduction degree and no further sintering, little higher than the reaction voltage was selected. And the remaining NiO phase is minor phase and the crystallite sizes are only slightly changed according to the XRD patterns.

Original comment 2:

On page 3, the authors state that the plasma has a “synergetic effect on the

physicochemical properties of catalysts. ” Although the concept of synergy (synergistic or synergetic) is used very loosely in the literature, it should only be applied to situations in which the adjustment of two variables together creates a much larger effect than changing one variable and then the other in sequence. That was not shown in this paper. I would call the effect of the plasma on the catalyst surface an “additional” effect.

Response:

“synergetic effect on the physicochemical properties of catalysts ” has been corrected to “additional” to avoid an ambiguous understand of the role of plasma that plays in CO₂ methanation.

Original comment 3:

On page 4 the authors should add detail about their synthetic method. For example, what were the concentrations of Ce, Ni and citric acid in their ethanol solution? Was the infiltrated template removed from the funnel during the drying stages, or was the entire funnel and template dried together? The line, “The hybrids were calcined at 550C for 6 h with a hold at 300C for 1 h at a ramp rate of 1 C/min ” is a bit confusing. I would suggest, “The temperature of the hybrids was raised to 300C at a rate of 1 C/min, held at 300C for 1 h, raised again at 1 C/min to 550C, and calcined at that temperature for 6 h. ”

Response:

More details about preparation of the catalyst are added as recommended in the context to improve the fidelity and avoid confuse. Typically, 5.04 g Ce(NO₃)₃·6H₂O, and 3.87 g Ni(NO₃)₂·6H₂O (molar ratio 1.15:1) were dissolve in 25.0 mL 95% ethanol solution with addition of equimolar citric acid (5.30g) as a complexing agent. And during the drying process, the infiltrated template was removed from the funnel and transferred to a vacuum oven. The comment on calcination process is helpful to avoid confuse.

Original comment 4:

On page 6, the authors should describe how they calibrated the GC to obtain mole fraction values.

Response:

The influence factor for each component was calibrated by 2 standard gas mixture through external standard method.

Original comment 5:

On page 8, the figure would be easier to read if the authors stacked the panels for the precursor template on the left and the panels for the catalysts on the right.

Response:

The figure has been rearranged to improve the readability.

Original comment 6:

On page 10 the authors claim that the crystallites of Ni and CeO₂ are smaller for the plasma treatment, but do not provide confidence intervals

Response:

More details and more interpretation are added to improve the fidelity and readability. In Table 1, the FWHM of Ni, NiO and CeO₂ differ significantly. And the Scherrer Equation are also cited as Equation 4. When processing the XRD patterns, all the peaks were carefully identified, fitted and refined to obtain all the profiles. And the residual error of fit was less than 5% for each catalyst. In this work, compared with catalyst after thermal reduction, all the components in the plasma assisted reduced catalyst shows smaller crystallite sizes and they are close to that of fresh catalyst.

There are 2 methods to obtain the crystallite size: calculation by Scherrer Equation based on XRD patterns and measurement and statistics from HRTEM images. In this work, the crystallite size is derived from Scherrer Equation. And through this method, the confidence interval cannot be given. The similar type of data can be seen in (Q. Pan *et al.*, 2014, *Catal Commun*, 45, 74-78), (T. A. Le *et al.*, 2017, *Catal. Today.*, 293-294, 89-96), (M. Nizio *et al.*, 2016, *Catal. Commun.*, 83, 14-17) and (M. Nizio *et al.*, 2016, *Int. J. Hydrogen Energ.*, 41, 11584-11592). Data derived from HRTEM images can

provide the clear confidence interval as shown in (W. Nie *et al.*, 2017, *Fuel*, 202, 135-143). The clear confidence interval cannot be given and the result is also reliable.

Original comment 7:

The authors should consider tweaking the legends in their figures to help readers who print the article in black and white: Organise the legends so that the top legend line matches the top trace, etc.

Response:

As shown in context, the figures have already been modified and rearranged to improve the readability of the manuscript.

References

1. Abelló, S., C. Berrueco and D. Montané (2013). "High-loaded nickel–alumina catalyst for direct CO₂ hydrogenation into synthetic natural gas (SNG)." *Fuel* **113**: 598-609.
2. Kogelschatz, U. (2003). "Dielectric-Barrier Discharges: Their History, Discharge Physics, and Industrial Applications." *Plasma Chemistry and Plasma Processing* **23**(1): 1-46.
3. Le, T. A., M. S. Kim, S. H. Lee, T. W. Kim and E. D. Park (2017). "CO and CO₂ methanation over supported Ni catalysts." *Catalysis Today* **293-294**: 89-96.
4. Liu, Q. and Y. Tian (2017). "One-pot synthesis of NiO/SBA-15 monolith catalyst with a three-dimensional framework for CO₂ methanation." *International Journal of Hydrogen Energy* **42**(17): 12295-12300.
5. Nie, W., X. Zou, X. Shang, X. Wang, W. Ding and X. Lu (2017). "CeO₂-assisted Ni nanocatalysts supported on mesoporous γ -Al₂O₃ for the production of synthetic natural gas." *Fuel* **202**: 135-143.
6. Nizio, M., A. Albarazi, S. Cavadias, J. Amouroux, M. E. Galvez and P. Da Costa (2016). "Hybrid plasma-catalytic methanation of CO₂ at low temperature over ceria

- zirconia supported Ni catalysts." International Journal of Hydrogen Energy **41**(27): 11584-11592.
7. Nizio, M., R. Benrabbah, M. Krzak, R. Debek, M. Motak, S. Cavadias, M. E. Gálvez and P. Da Costa (2016). "Low temperature hybrid plasma-catalytic methanation over Ni-Ce-Zr hydrotalcite-derived catalysts." Catalysis Communications **83**: 14-17.
 8. Pan, Q., J. Peng, T. Sun, S. Wang and S. Wang (2014). "Insight into the reaction route of CO₂ methanation: Promotion effect of medium basic sites." Catalysis Communications **45**: 74-78.
 9. Parastaev, A., W. F. L. M. Hoeben, B. E. J. M. van Heesch, N. Kosinov and E. J. M. Hensen (2018). "Temperature-programmed plasma surface reaction: An approach to determine plasma-catalytic performance." Applied Catalysis B: Environmental **239**: 168-177.
 10. Petitpas, G., J. Rollier, A. Darmon, J. Gonzalezaguilar, R. Metkemeijer and L. Fulcheri (2007). "A comparative study of non-thermal plasma assisted reforming technologies." International Journal of Hydrogen Energy **32**(14): 2848-2867.
 11. Tu, X., H. J. Gallon and J. C. Whitehead (2013). "Plasma-assisted reduction of a NiO/Al₂O₃ catalyst in atmospheric pressure H₂/Ar dielectric barrier discharge." Catalysis Today **211**: 120-125.
 12. Zhang, X., W.-j. Sun and W. Chu (2013). "Effect of glow discharge plasma treatment on the performance of Ni/SiO₂ catalyst in CO₂ methanation." Journal of Fuel Chemistry and Technology **41**(1): 96-101.
 13. Zhou, G., H. Liu, K. Cui, H. Xie, Z. Jiao, G. Zhang, K. Xiong and X. Zheng (2017). "Methanation of carbon dioxide over Ni/CeO₂ catalysts: Effects of support CeO₂ structure." International Journal of Hydrogen Energy **42**(25): 16108-16117.

Appendix B

Responses to Referee' comments

Comments to the Author(s)

The authors have addressed the minor points I made in my previous review; however, I don't believe that they understood the point I was trying to make in what I describe as my "main scientific issue."

It seems to me that the two research questions this paper is mean to address are (1) whether low temperature plasma is the better method for reductively activating the Ni catalysts described and (2) whether plasma is a better method than thermal for methanation. I believe that the latter is better answered than the former because the same catalyst was used in the methanation experiments, so that was controlled, and the conditions (flow rate and temperature) varied during those experiments, so there was a kind of optimization built in to the methanation experiment.

On the other hand, the only thing the activation experiments demonstrate is that under two different reduction conditions, two different catalysts can be produced from the same starting material. The data do not show that plasma is the better method. The authors have provided references and data to show that their choice of thermal reduction conditions (100 mL/min H₂, 470C, 60 min.) were appropriate to activate the catalyst. That was never in dispute: The thermally-activated catalyst had reasonable activity, although lower than the plasma-activated one. My point is that the question of whether plasma is—in general—the better method for activating these Ni catalysts is not answered, just because one particular plasma-activated catalyst outperformed one particular thermally-activated catalyst.

So what would answer the question? I believe that if the authors attempted to optimise the conditions for the thermally-activated catalyst (reaction time, hydrogen flow and reduction temperature), and set their very best thermally-activated catalyst against their one plasma-activated catalyst, that would be sufficiently convincing. If their un-optimized plasma catalyst can beat their optimized thermal catalyst, I would have no argument.

Response:

We regretted for the not good understanding and response for the concerned "main scientific issue" in last time, and thanks for the clear explanation and instruction for how to make a clarification on this question.

Yes, we wanted to give two research clues in our paper 1) plasma reduction is a

better method for the catalyst's low temperature activity, and 2) plasma reaction is a better method than thermal for methanation. Just as the reviewer's comments, the question (2) is better answered, because the tests were performed on the same catalyst with the same state. However, for the question (1), the logical relationship constructed in the manuscript is not firm and convincing enough. The indispensable step is optimizing or screening the thermal-reduction condition and then compares them with the case of plasma reduction.

According to the reviewer's suggestion, we carried out the thermal reduction at different temperature 350 °C, 400 °C and 450 °C with slower temperature-rise step 2 °C /min. And then the activity-test experiments were performed on each catalyst, the furnace temperature was raised slowly to avoid temperature-runaway. The XRD and TPD spectrum were also tested for each catalyst.

Fig. 1 XRD spectrum of Ni-Ce catalysts with different treatment condition

Fig. 2 CO₂ TPD spectrum of Ni-Ce catalysts reduced under different condition

Fig. 1 shows the XRD spectrum of catalysts with different treatment condition. Plasma reduction and 450 °C thermal reduction give clear and sharp Ni peaks, while there are only irregular and small drum-bags at the positions for the catalysts reduced under 350 °C and 400 °C. This indicates that NiO is reduced slowly and not form the regular Ni crystal phase under lower thermal temperatures. The CO₂-TPD tests on the catalysts are shown in Fig.2. The order of CO₂ desorption peak area is: 350 °C reduction \approx 400 °C reduction < 450 °C reduction < plasma reduction. However, we failed to calculate the crystal size for the catalysts reduced under 350 °C and 400 °C, because of irregular broad peaks.

The activity test results are shown in Fig. 3. It is obvious that the plasma reduced catalyst gives higher low-temperature activity and CO₂ conversion than those reduced by thermal method. The CO₂ conversion reached ca.87% at 300 °C (catalyst bed temperature) over plasma reduced catalyst, while for those thermal cases a higher temperature 370 °C (catalyst bed temperature) is needed to reach the maximum CO₂ conversion (79-81%). The maximum CO₂ conversion over 450 °C reduced catalyst is a little higher than that of 350 °C and 400 °C reduced catalysts. And judged from Fig. 3 a, the best thermal reduction temperature should be around 450 °C.

Figure 3. Activity test curves of 3DOM Ni-Ce catalyst reduced under different conditions, a) CO₂ conversion vs furnace temperature, b) CO₂ conversion vs catalyst bed temperature.

In general, the supplementary experiments proved the first question “plasma reduction is a better method than thermal reduction”. Plasma reduction led to higher CO₂ adsorption capacity, higher low-temperature activity and higher CO₂ conversion. And we supplemented these optimized process and revised the manuscript according to the new experiment results, as highlighted by yellow mark.